# Physiological Factors Linking Insecure Attachment to Psychopathology: A Systematic Review

**DOI:** 10.3390/brainsci11111477

**Published:** 2021-11-08

**Authors:** Marta Tironi, Simone Charpentier Mora, Donatella Cavanna, Jessica L. Borelli, Fabiola Bizzi

**Affiliations:** 1Department of Educational Science, University of Genoa, Corso Podestà 2, 16128 Genoa, Italy; marta.tironi@edu.unige.it (M.T.); simone.charpentiermora@edu.unige.it (S.C.M.); donatella.cavanna@unige.it (D.C.); 2Department of Psychology and Social Behavior, University of California, Irvine, CA 92697, USA; jessica.borelli@uci.edu

**Keywords:** attachment, psychopathology, emotion dysregulation, physiological parameters, skin conductance level, cardiac slowing, respiratory sinus arrhythmia, moderation or mediational analysis

## Abstract

Although many studies have documented associations between insecure attachment and psychopathology, attachment may not confer risk for psychopathology independently, but rather through its interaction with emotional, social, and biological factors. Understanding the variables through which attachment may lead to psychopathology is therefore important. Within this domain of research, the role of physiological factors is poorly investigated. What are the relevant domains and why, when, or for whom do they influence mental disorders relating to attachment? The current systematic review aims to answer these questions. Results reveal that physiological indices of emotional regulation play a role in explaining and/or determining the relationship between attachment and psychopathology. Specifically: (1) combined with insecure attachment, higher skin conductance level (SCL), lower cardiac slowing, and respiratory sinus arrhythmia modulation (RSA) contribute to different psychopathological indicators and behavioral/psychological dysfunctions, although the latter predicts a contradictory pattern of findings; (2) insecure-avoidant attachment is more consistently linked with stress and emotional dysregulation when combined with RSA, while anxious attachment confers risk of depressive symptoms when combined with SCL. We concluded our discussion of the results of seven studies by outlining a plan to move the field forward. We discuss the quality of the assessment, methodological limitations, and future directions, highlighting the need to extend the research to clinical samples.

## 1. Introduction

Beginning with Bowlby’s conceptualization [1,2], attachment is defined as a complex psychobiological system in which the infant attempts to maintain proximity with a primary caregiver in order to cope with dangers and threats and ensure the child’s psychological and physical survival. Attachment is a construct that underlies the functioning of different processes (neural, physiological, and behavioral) that contribute to maintaining a ‘vital biological function’ [3]. Individual differences in how a child responds biologically to positive or negative stimuli in the environment help to define his or her developmental trajectories [4] and should be prognostic of subsequent adaptation [5]. Attachment may be secure, insecure, or disorganized. Insecure attachment is further subdivided into avoidant attachment, which is characterized by the avoidance of negative emotions and the dismissal of dependency on attachment relationships, and anxious attachment, characterized by a hyper-activation of attachment-related needs and emotions [6]. It has long been established that insecure attachment (i.e., as a result of a history of relational experiences with inconsistent parental responses) and disorganization (i.e., due to the exposure to highly frightening experiences within the attachment relationship) [6,7] are related to developmental dysfunction and mental disorders, but the magnitude of this relationship is relatively modest, especially when compared to the strength of theorizing around this relation (e.g., [8,9]). In addition, for a few years now, research regarding the psychobiology of attachment has increased significantly, defining correspondences between attachment insecurity and biobehavioral dysregulation (such as the stress regulatory system) (e.g., [10]). Our review aims to integrate these lines of research to investigate the relationships of this complex psychobiological model with different aspects related to psychopathology (e.g., psychopathological indicators, behavioral/psychological dysfunctions, specific diagnosis).

### 1.1. Physiological Factors Linking Attachment and Psychopathology

Following the World Health Organization’s definition (ICD-10; 1992, [11], p. 5), mental disorder implies “a clinically recognizable set of symptoms or behavior associated in most cases with distress and with interference with personal functions”. The American Psychiatric Association (DSM-IV, 1994, [12], pp. xxi–xxii) also defines psychopathology, synonymous with mental disorder itself according to APA dictionary, as a “clinically significant behavioral or psychological syndrome or pattern that occurs in an individual and that is associated with present distress (e.g., a painful symptom) or disability (i.e., impairment in one or more important areas of functioning) or with a significantly increased risk of suffering death, pain, disability, or an important loss of freedom” and a criterion for disorder is that “the disturbance causes clinically significant distress or impairment in social, occupational, or other important areas of functioning”. Following these definitions, any behavior, emotion, or experience which causes impairment, distress, or disability can be considered psychopathological.

Attachment insecurity is associated with mental health difficulties in childhood and adolescence, including internalizing [13,14,15] and externalizing symptoms [16,17], as well as in adulthood, when attachment insecurity is associated with psychopathology more generally [18,19] and with depressive symptoms in particular [20]. There is a significant continuity of attachment from infancy to adulthood [21]; much research supports the connection between attachment in childhood and the formation of specific developmental trajectories [22,23], with attachment influencing a variety of factors from emotion regulation strategies [24,25] to adult personality [26].

It is important to note that, following the principle of multifinality [27], the influence of any one risk factor (individual, biological, environmental) depends on when it occurs and how it interacts with other factors [28]. 

Therefore, defining which mechanisms and conditions of physiological factors explain the association between attachment and psychopathology allows us to answer our primary question, namely how, why, and/or for whom this association occurs. A vital step in this process entails the analysis of mediator variables, which intervene to explain the indirect link between attachment and psychopathology and moderator variables, which define the conditions of this link. Moderator and mediator variables of the association between attachment and psychopathology have been investigated in some reviews and meta-analysis including for ADHD symptomatology [29]; for eating symptoms [30]; for depression [31,32]; for post-traumatic stress, [33] and in several empirical studies (i.e., [34,35]) as well as in different cultural settings [36]. 

### 1.2. Physiological Factors as Emotion Regulation Index

One potential moderator or mediator of interest is physiological reactivity to stress, which is an index of emotion reactivity and regulation. For instance, attachment anxiety and avoidance are associated with altered physiological reactivity to stress, which in turn is linked to deficits in emotional regulation [37]. According to Spangler and Zimmerman (1999) [38], the attachment system acts as a regulatory mechanism in the interaction between the different physiological factors underlying emotions, influencing both the perception of emotion and coherence in communicating emotions. Respondents with secure attachment will therefore have greater regulatory capacities (for an example in the context of infancy, see Spangler and Schieche, 1998 [39]). Several researchers have presented models linking affective dysregulation and mental disorders, arguing that mental health depends crucially on affective states such as emotions, stress responses, and state of mind [40,41,42]. Indeed, self-regulation is an early ability to exert control over cognition, behavior and emotions, especially negative ones, and does not appear to confer specific risk for single disorders but rather to present general risk for multiple disorders. The inadequate development of this ability may become a transdiagnostic risk factor for the appearance of psychiatric disorders [43,44], thereby having an impact on social adaptation throughout development [45,46,47]. Bradley [40] also presented a model in which affect dysregulation is a core factor in different types of psychopathologies, calling it a “general arousal factor”. Moreover, impulse regulation is a relevant indicator of the P-axis in the PDM-2, contributing to the assessment of personality organization [48]. Effective emotional regulation strategies develop precisely in the context of security with attachment figures [49]. When this does not occur, insecure attachment and poor regulation correlates with psychopathological trajectories and maladjustment throughout life [50,51]. For this reason, the intention of this review is not to focus only on specific mental disorders but to understand psychopathology more broadly.

The links between emotion regulation and psychopathology can also be found in the relational nature of primary bonds with caregivers and peers. Sroufe et al. (2000) [52] define relational problems as ‘criteria for disorder’ (p. 76), arguing that even the nosographic-descriptive diagnostic systems (i.e., DSM-IV [12]) highlight the centrality of such indicators as detectors of the main mental disorders, both in childhood and adulthood. For example, during childhood, the massive presence of problems in the peer group (e.g., rejection, exclusion, withdrawal, internalizing problems) can lead to disorder, whereas peer competence acts as a protective factor for behavioral deficits and is associated with the absence of psychopathology [52], configuring relationships both as a diagnostic criterion and as a context and risk factor for psychopathology. Emotional regulation and peer relations are, moreover, closely linked and are identified as a pathway for psychopathology in childhood and adolescence [53].

### 1.3. Which Physiological Factors of Emotions Are Involved in Research?

Studies in neuroscience and physiology have led to the realization that the psychophysiological correlates of emotions might influence mental health (e.g., [54,55]). Within this broad literature on physiology as an index of emotion regulation, there are two main groups of physiological factors that have primarily been considered. The first one is heart rate variability (HRV), indexed by respiratory sinus arrhythmia (RSA), i.e., the heart rate variability synchronized with respiration [56], and cardiac slowing (CS). Resting RSA is the physiological signal of vagal tone, an indicator of parasympathetic nervous system (PNS) functioning. RSA and HRV are valid and reliable transdiagnostic biomarkers of mental disorders and emotion regulation in humans [57,58], e.g., reduced tonic RSA and also excessive RSA reactivity to emotion is linked to poor emotion regulation and psychopathology [59]. Autonomic swings seem to be more plastic, faster, and easier to disengage when driven by regulation in the PNS compared to those in sympathetic nervous system (SNS) [60]. Porges et al. [61,62,63] report that individuals with high vagal tone, hence a PNS that is functioning well, seem to respond more rapidly and flexibly to stressful tasks and are able to recover more effectively from increased arousal in a stressful situation. On the contrary, low vagal tone has been linked to poor emotional control in infants [61], but also to higher levels of depression, anxiety, and anger in adults and children [64,65]. It is also connected with maladjustment [66,67]. Diamond and Hicks (2004) [68] suggest that individual differences in vagal tone partially explain attachment-related individual differences in handling emotions, since emotion regulation has been both empirically and theoretically associated with insecure attachment [51,69,70]. Children with lower vagal tone have worse behavioral outcomes in association with lower-quality parenting [71,72].

Cardiac slowing (CS) is a transient measure, indicative of a neural system that includes the anterior cingulate cortex and involved in both the perception of physical pain and social rejection [73]. Its assessment uses the rapid action of the vagus nerve that overcomes cardiac sympathetic innervation for up to 2 s, inducing it only under parasympathetic influence [74]. It is associated with negative feedback (e.g., error feedback) and it correlates with expectancy violation (e.g., [75,76]). In school age children, higher cardiac slowing is associated with peer rejection [77]. In research on individual differences among adults with neuroticism and depressive symptoms, cardiac deceleration was greater in the face of unexpected negative judgment [78]_._

The second factor is electrodermal activity (EDA), indexed by skin conductance level (SCL), a non-invasive method of assessing the sympathetic branch activity (SNS) of the autonomic nervous system. SCL is a background tonic component of EDA, providing a measure of the autonomic changes in the electrical properties of the skin [79]. Previous research has shown that SCL increases during specific social tasks [80], while reduced electrodermal activity is associated with lower SNS influence [81]. As interest increased in studying ANS response patterns as correlates of emotion, research on SCL has also bloomed. High SCL has been associated with stress and anxiety and a decrease in parasympathetic indices such as vagal tone (see [82]). Prior work investigated the moderating role of SCL variability in the relationship between different constructs, showing that relationships among variables vary by SCL (e.g., between parental depressive symptoms and psychological adjustment [83]; between parenting style and child externalizing behavior [84]; between parental psychological control and relational aggression in young adulthood [85]; and between family conflict and disordered eating behaviors [86]).

Despite the large number of studies examining links between psychophysiology and mental disorders, there is a lack of studies examining physiological factors as moderator or mediator variables in the association between attachment and psychopathology. The main contribution of this review is to answer the following question: which physiological factors link attachment to psychopathology in the general population, and why and for whom do they operate over the lifespan?

## 2. Methods

This review was prepared according to the preferred reporting items for systematic reviews and meta-analyses (PRISMA) recommendations. Due to a high degree of heterogeneity in the studies, we referred to the synthesis without meta-analysis in systematic reviews checklist (SwiM [87]) as a complement and extension to the PRISMA protocol (Appendix A, Table A1). This review has been pre-registered on PROSPERO.

We followed these steps:Selection of our topic of interest (identifying the physiological factors underlying the relationship between attachment and psychopathology and how these work);Literature search and screening of results;Summary of results and discussion;Conclusions.

The results of the mediation or moderation analysis to evaluate the studied relationship were extracted from the studies and summarized in the discussion.

### 2.1. Data Collection

We identified two strategies to collect studies for possible inclusion. First, a systematic literature search was conducted between April and May 2021 (last search on 20 May 2021) through research on the following databases: Web of Science, PubMed, PsycInfo, PsycArticles, Scopus, plus Google Scholar and ProQuest to identify works from other sources. We use the same investigation strategy for all databases. The search included the following key terms (see Table 1), combined with the Boolean vector AND, to capture titles, abstracts, or keywords along the full text.

The choice of using such broad key words to investigate psychopathology seemed useful to include both established disorders and symptoms (DSM oriented) but also behavioral or psychological dysfunction (per APA and WHO definitions), following the logic that by including the macro-categories related to psychopathology (e.g., ‘mental disorder’ or ‘psychiatric’), it would also be possible to detect both specific dysfunctions (e.g., peer-problems, negative affectivity, emotional dysregulation) and specific psychopathological indicators (e.g., anxiety or depression symptoms).

The search took into account articles without limiting the year of publication, while it considered only the results published in English and Italian and only peer-reviewed articles (see Figure 1).

Second, we found further potentially relevant articles by screening the reference lists through the literature search.

### 2.2. Selection Criteria and Eligibility

The PICOS approach was used to clarify the questions of this review and to select a priori the inclusion and exclusion criteria of the studies. After removing duplicates using Zotero, a total of 3549 potentially relevant abstracts were assessed. The first author screened all the studies and carried out the study selection. Doubts, discrepancies, and the final selection were discussed in several meetings with other authors. Two authors, working independently, assessed eligibility based on the inclusion criteria in Table 2. 

### 2.3. Quality Assessment

To assess the methodological quality of studies included in this review, a modified checklist of eight items derived from Garcia et al. (2019) [30] was used to identify the risk of bias (Appendix B, Table A2). The items included: (a) clear description of objectives; (b) appropriate design; (c) representative sample; (d) psychometric characteristics of the mediator/moderator and outcome variables reported; (e) significance of predictor on a mediator, and mediator on the outcome variable; (f) interaction test found in moderation (as an alternative to the previous point, depending on the analysis used), (g) findings clearly described; and (h) control for confounds. 

Quality levels of evidence were found in a 8-point scale (score 1—“yes” or 0—“no” for each item) that provides a rating for the quality level (8 = strong; 7–5 = moderate; 4–0 = weak; see Table 3). The studies were rated by two authors independently. Disagreements were solved by a discussion with the third author.

## 3. Results

### 3.1. Main Characteristics of Studies Included

The low number of papers selected (*n* = 7; [68,88,89,90,91,92,93]) does not allow us to extend or generalize results or findings. However, we can draw a great deal of information from these studies to underscore some meaningful features and outline an agenda to move the field forward.

Table 4 shows the characteristics of the seven studies regarding the study design (cross-sectional *n* = 4 and longitudinal *n* = 3, so causality in the associations between variables cannot be inferred); the statistical analysis (moderation *n* = 4, mediation *n* = 3); the country and language (only Western countries, only English). The sample ages are heterogeneous (children *n* = 4, adults *n* = 3), but we chose to include all ages in any case since there is a significant continuity in attachment from infancy to adulthood [21]. Studies included both genders, except one which employed only men, due to documented gender differences in responses to their experimental manipulations (see [68]). The sample size ranged from 60 to 213 participants and the mean age of participants across studies ranged from 20.99 weeks to 40.1 years, but in Murdock et al. [92] the composition of the sample is not made explicit. All studies used non-clinical samples.

Attachment was assessed using self-report measures in almost all studies except in White et al. [93] which used a narrative test and in Conradt et al. [89] which used an experimental procedure. In terms of the attachment figures assessed as the focal points of the studies, the studies assessed romantic partners (*n* = 3), mothers (*n* = 3), and mothers and fathers (*n* = 1).

The mediator/moderator variables chosen in this systematic review referred to many physiological factors detectable in literature. Three indicators are found in the studies reviewed. The most represented is the respiratory sinus arrhythmia (RSA) (*n* = 5), both during a baseline and an activated task. The other cardiac parameter was the event-related transient cardiac slowing (CS) (*n* = 1). The last factor detected was skin conductance level (SCL), assessed by some electrodes on fingers in participants’ non-dominant hands.

Several experimental tasks were used to detect physiological factors (see Table 4). 

Psychopathology included a large variety of behavioral and psychological dysfunctions and psychopathological indicators as defined above. Nevertheless, in our systematic research, not only psychopathological indicators (depressive symptoms *n* = 1) but also behavioral and psychological dysfunctions (negative affectivity and self-regulation = 3, perceived stress n = 1; peer and behavioral problems *n* = 2) were found. Lastly, no specific disorders (i.e., DSM diagnosis), were found. This will be discussed in the Section 4. The seven studies had a limited range of quality. Their scores ranged from 7 (moderate) to 8 (strong) in the methodological quality assessment (see Table 3 for details and Table 4 for individual study quality assessment).

### 3.2. Main Results

This review analysis follows a narrative and thematic synthesis. To explain why and/or for whom attachment links to the psychopathology, we discuss the results by presenting the physiological factors separately to highlight the different processes involved in the relationship studied.

#### 3.2.1. Group 1: Respiratory Sinus Arrhythmia (RSA; *n* = 4 Studies)

RSA is the most represented factor. Regarding moderation analysis, in Sbarra and Borelli [90], highly avoidant participants who had increases in RSA during the divorce-related mental activation task (DMAT) showed improvements in a measure of self-regulation, the self-concept reorganization (i.e., revise their basic sense of self after a breakup; [90]) three months later, while highly avoidant adults who had a decrease in RSA showed relatively little self-concept reorganization. No association between attachment anxiety and worsening of self-concept disturbance was found. In Fagundes et al. [88], the association between attachment avoidance and poor current adjustment to loss was positive for those with fewer stress-induced changes in RSA, whereas this connection was negative for those with higher levels of stress-induced changes in RSA, who were better able to regulate their emotions after the grief. Attachment anxiety was positively associated with poor loss adjustment, but no moderation effect was found. Diamond et al. [68] explored two different independent variables: attachment (with ECR) and attachment-related functions (with WHOTO). The indirect relationship between experiencing a high perception of emotional safety within a current attachment relationship and more effective recovery from emotion, after laboratory-induced anger in adulthood, was mediated by baseline RSA. The same indirect effect was not found with anxious attachment, while there was only a significant direct relationship between anxious attachment and a decrease in resting RSA. No significant effects for avoidant attachment.

On the contrary, in the last two studies, high levels of stress-induced RSA mediated or moderated worsening outcomes. In Murdock et al. [92] adult attachment avoidance was indirectly associated with self-reported health via general stress, but only among those with high stress-induced RSA, finding a moderated-mediation model. Attachment anxiety was directly associated with general stress and health without moderation. Conradt et al. [89] found similar results in infancy, where high baseline RSA was a discriminant factor as well. Infants raised in disorganized environments showed an increased level of behavioral problems at 17 months, while those who benefited from a safety-promoting environment (secure attachment) showed a decrease. 

#### 3.2.2. Group 2: Skin Conductance Level Variability (SCLV; *n* = 1 Study)

In Bosmans et al. [91], SCL variability moderated the associations between attachment anxiety and depressive symptoms, such that only when children showed significantly increased SCLV (i.e., 0.05 DS above the mean or more) this association was significant. This result was found in 34% of the children involved. Attachment avoidance was related to depressive symptoms but was not a significant predictor if in interaction with SCLV.

#### 3.2.3. Group 3: Cardiac Slowing (CS; *n* = 1 Study)

Heart rate variability mediated the association between perception of security in current attachment relationship and peer-problems [93]. When parental representations were positive these had an effect on problems with pairs via cardiac deceleration. The better these representations were at pre-school age, the fewer peer problems were revealed during school age, and this relationship was mediated by cardiac slowing. Children with less positive representations of their parents did not show a significant change in cardiac slowing during the rejection events in the experimental task (being excluded from receiving the ball), in contrast to children with more positive representations, in which in turn Cardiac slowing is positively linked to peer problems.

## 4. Discussion

This study aimed to investigate the relationships between attachment and psychopathology exploring influencing variables that mediate or moderate this connection. We focused on physiological factors as indices of emotional regulation [57,63,111] to explore this goal. In general, the results show that physiological factors are useful in explaining why, when, or for whom insecure attachment related to different aspects of psychopathology and at what levels of physiological variables this relationship may occur. However, a reflection on the definition of the outcomes found is necessary.

Based on the above-mentioned connecting physiology, emotional regulation and mental disorder, attachment seems to be associated with a broad spectrum of symptomatology (psychopathological indicators such as depression symptoms) and risk factors for mental disorders, which include a wider scale of behavioral, emotional, and regulatory dysfunctions (e.g., poor self-regulation, perceived stress and so on), rather than discrete disorders. Some of these variables allow us to question the meaning of psychopathology and draw attention to “blurring the border between pathology and variants of the ‘normal’” [112]. In recent years, the literature has increasingly moved towards investigating the composite nature of psychopathology, attempting to shift beyond diagnostic classification in search of transdiagnostic risk factors that reflect the nature of comorbidity [113]. Focusing on studies with specific disorders has less ecological validity and may also have less power to detect associations between risk factors and psychopathology [114]. In a recent review of a population of young people, Lynch et al. [115] highlighted that stressful life events, peer and friendship problems, low effort control and negative affectivity, among others, increase transdiagnostic risk and are correlated with a general psychopathology factor (or ‘p’ factor, [116]) that seems to reflect a shared vulnerability between different mental disorders [117]. In the light of these reflections and of the results found, in retrospect we suggest it would be better to consider also psychopathological risk factors as useful indicators as prodromes of mental disorder. These risk factors, maybe more than overt symptoms or disorders, would seem to be useful indicators for the outcomes’ evaluation in the relationship between attachment and physiology. The following discussion should be interpreted in the light of these considerations.

Starting from the RSA, when avoidant attachment is high, low RSA, or its reduction or inhibition, negatively moderates emotional regulation in separations and loss in adolescents and adults [82,84]. On the contrary, two of the studies considered here [89,92] diverge completely on the direction of RSA and vagal tone (high or low) in relation to positive or negative consequences related to attachment. These latter two studies find that in both adults and infants with high baseline RSA, there was a significant difference in psychopathological indicators (self-reported or detected by caregivers) depending on attachment classification. In the literature overall there are different patterns of results with RSA in different studies and it is a topic which is highly disputed. This contrast in ratings on the same measure may be explained by several factors, for example it might depend on the different experimental tasks or the degree of RSA suppression. It is important to note that both latter studies involved participants with low-income families, and it was a determining factor in their developmental trajectories. In line with previous literature [118], one explanation could be found in the discrepancy of income in the samples of the different studies, although we do not have enough data to state this with certainty. Heterogeneity of results may also reflect different timing of assessments. In addition, the outcomes considered were dissimilar: studies with similar variables (e.g., loss [88,90]) achieved similar trajectories in moderation patterns, the others could not be compared. The experimental tasks used could not be compared too; some used social stress situations, while other studies acted on individual thoughts or feelings.

In summary, in all these studies, RSA modified the association between attachment and affective regulation, indicating that it is an important individual factor in determining how individuals regulate negative emotions they may experience in relying on others in interpersonal relationships. However, the effectiveness of its direction needs to be clarified. With respect to attachment patterns, our review highlights a significant pathway in which individuals high in avoidant attachment appear to be most vulnerable to negative outcomes depending on self-regulatory abilities indexed by RSA [88,90,92]. This finding is in line with other work that linked attachment avoidance, but not anxious attachment, with worse outcomes—such as post-cancer treatment quality of life [119]—in interaction with stress-induced RSA. The latter has been operationalized as self-regulatory effort/strength [120]. For this reason, the research and our results emphasize that those who show avoidant attachment are more vulnerable to poor emotional regulation only if this is underpinned by low self-regulation strength. The strategy that individuals with high attachment avoidance are likely to use involves shifting attention away from potential threat signals and attenuate experiences of negative affect [69,121]. Its physiological correlate is greater withdrawal of the vagal brake [122].

CS has previously been correlated with the breach of expectation [75]. The exclusion given by the experimental task comes as a greater surprise, so the physiological system is more activated in children with positive parental representations. This can be interpreted in two ways: first, some argue that cumulative experiences of sensitive and responsive relationships give rise to greater positive expectations of inclusion, or that children who are more secure in their attachment relationships show less inhibition in processing negative social information [93]. In the literature, however, it emerges that cardiac slowing is associated with negative outcomes [123]. For children who consistently experience fewer positive interactions with their care environment, negative social information may represent a greater risk factor in developmental trajectories, leading them to minimize negative feedback from the environment by deactivating their response [124]. Future research is needed to better explain these data, perhaps considering whether different attachment strategies are associated with unique physiological responses to dysfunctional circumstances. 

Moving to electrodermal activity, Bosmans et al. [91] emphasized its role in showing depressive symptoms during childhood based on attachment anxiety, and this is in line with previous works suggesting that perceived stress is a risk factor for the onset of depressive symptoms [125], where greater SCL variability influences the relationship between environmental impact and symptomatology [126]. In contrast, the non-significant effects of avoidant attachment could be explained by the high correlations with SCLV. Additionally, avoidant attachment may act as a “protective” factor, together with the deactivation of emotional regulation, for the development of emotional problems [127].

In summary, it appears that under conditions of attachment vulnerability, performance in physiological regulation has a negative influence on mental health.

However, research in this field is still in its infancy.

### 4.1. Risk of Bias in Included Studies

As far as the systematic review is concerned, we may have incurred publication bias, as we have only selected peer-reviewed studies published in peer-reviewed journals. On one side, this guarantees that we are utilizing studies of a certain quality, but on the other side, pre-print research or unpublished data are not captured, with the risk that only partial studies with significant results were included in the review [128]. Another possible weakness could lie in the choice of keywords: by eliminating the search string linked to mediation and moderation, we could perhaps have obtained a greater number of eligible studies. However, we decided to restrict the field to only studies that used mediation or moderation analyses in the hope of answering our research question (i.e., What are the physiological factors and why or how do they influence the different aspects of psychopathology relating to attachment?) more accurately and increasing the methodological quality of the reviewed studies. Regarding psychopathological key-words, we suggest including key-words on psychopathological risk factors (e.g., ‘stressful life events’ or ‘negative affectivity’) in future reviews or meta-analysis intent on studying the same relationships between constructs.

As we expected to find few studies on our topic, we considered a wide age range, from infancy to adulthood, which precluded us from focusing on specific age groups.

A final limitation concerns the qualitative assessment of the studies as the instrument used is an adaptation of a checklist based on mediation studies. Therefore, the score should only be considered as a guide.

Regarding risks of bias in the selected studies, some limitations can be found especially in terms of sample size and the extreme heterogeneity of physiological assessment methods, which could have partially contributed to conflicting results, as well as the control of intervening variables.

Regarding the choice of attachment measures, except in two cases, only self-report questionnaires were used. First, this kind of measure may incur retrospective bias and distort results since participants may respond according to the principle of social desirability [129]. Consistent with previous works [130,131], these measures have limitations, especially when correlated only with other questionnaires. One of the major limitations is that the self-report used in the works reviewed only capture individuals’ subjective perceptions but fail to capture attachment disorganization, an essential factor related to psychopathology [132,133,134,135], which is possible with other measures such as narrative-observational (e.g., the Strange Situation Procedure [98]; the Child Attachment Interview [136]; the Adult Attachment Interview, [137]).

### 4.2. Future Research Agenda in Light of Limitations

It is not possible to draw definitive inferences from our work due to several factors: the paucity of studies considered, relatively low sample sizes, ages and conditions heterogeneity of participants, and the total absence of studies on clinical samples, especially among children. For this reason, our review stands out as a preliminary work that serves to highlight what is missing within the literature and to define how the research within this area might proceed. Accordingly, we provide some recommendations designed to help shape future research in this area by addressing some limitations this work has brought to light.

#### 4.2.1. Clinical Samples and Psychopathology

All the studies considered used non-clinical samples; therefore, the results refer only to the general population. This may be one of the reasons why the outcomes were not homogeneous. This highlights the emergent nature of the research which, however, limits the generalizability of the results. More studies with clinical samples may bring new data to the discussion. To enrich the debate, future research should therefore focus on the creation of case-control research designs in which the atypical population with a priori diagnosis is also considered. 

#### 4.2.2. Valid and Reliable Methods to Detect Physiology

Literature on neurophysiological correlates of attachment has flourished in recent years, but much of this work has focused, for example, on Gene x Environment research designs to determine what genetic factors influence pathways from insecure attachment to psychopathology [138,139] or has used endocrine mediators [140]. Assuming that both early attachment experiences and current attachments influence emotional regulation skills [49] and that emotional regulation is closely linked to psychopathological occurrence [141,142], creating a multi-layered search field from outside to inside, future research should more fully explore the connection with physiological measures, psychopathology and risk factors. Besides, it would be important to assess which individuals with clear psychopathological outcomes differ in physiological measures and attachment patterns. Further work using SCL and HR (heart rate) is needed. The latter is underrepresented in these reviewed studies but widely used in literature linked to attachment and mental disorders [143,144,145]. However, future research should consider recent developments that cast doubt on the validity of this measure to detect cardiac sympathetic modulation of the heart, especially in study tasks where verbal responses are involved, that could induce such large changes as to invalidate the power of the measure [146].

#### 4.2.3. Methodological Improvements (Sample and Statistical Analysis)

In order to improve the internal validity of studies, it is important for researchers to carry out a priori power analysis and then to explain and justify the choice of sample size [147]. As in the studies selected here, several observations are often stated, but not justified. In addition, it would also be desirable to select the participants randomly and to widen the pool to include people from different cultures, for better results generalizability. Statistical techniques should be improved too, such as by applying power-enhancing analyses such as SEM with the bootstrapping method or mixed models [148,149,150]. Furthermore, to enhance quantitative assessment, we propose to use the effect size measure with confidence intervals rather than *p*-values in the results reporting. Besides, the control of latent variables and intervening effects will then have to be better managed to increase methodological quality [151]. Finally, given the paucity of studies, it will be useful to ascertain mediation or moderation effects according to age distributions, perhaps including more longitudinal or follow-up studies.

#### 4.2.4. Diathesis-Stress vs. Differential Susceptibility

Some of the included studies appear to support the differential susceptibility hypothesis [152,153] in which vulnerable children are more susceptible to positive care environments than peers raised in safer conditions [4,153,154,155]. In Conradt et al. [89], increased physiological susceptibility appears to act as a protective factor in children raised in poorer environments, by making them more sensitized and influenced by their relationship with a responsive and sensitive caregiver. Following this hypothesis, Bosmans et al. [91] invited researchers to investigate not only dysfunction in children with high attachment anxiety and high SCLV, but also whether individuals low on attachment anxiety but with high SCLV are more likely to develop positive outcomes, under certain environmental conditions, assuming that SCLV could be a differential susceptibility factor [156]. Further research is needed to investigate this model, which contrasts with the diathesis-stress model [157,158]. The question is not trivial. The idea that insecure attachment strategies are interpreted as adaptive reactions to unresponsive environments rather than psychopathological symptoms [159] might be supported by these data. 

### 4.3. Clinical Implications

In the absence of clinical data to draw on, we briefly highlight the importance of prevention programs that consider and then work on emotional regulation and its physiological correlates, especially in childhood where developmental trajectories are more sensitive. There is evidence in the literature of how the early rearing environment influences young children’s stress physiology and emerging behavioral problems [160]. We suggest that researchers implementing evidence-based attachment programs could assess affective regulation, with its psychophysiological correlates, as an intervention target and use it as a positive prognostic dimension for psychopathological outcomes, planning to work with couples, parents, or individuals as well. Finally, studies by Conradt et al. [89] and Murdock et al. [92] highlight how children raised in poverty exhibit physiological traits susceptible to the characteristics of their care environment differentially compared to those living in less compromised environments. Expanding research on psychosocial status as an intervening variable in the relationship between attachment and psychopathology could open new avenues for evidence-based attachment programs in at-risk settings [161].

## Figures and Tables

**Figure 1 brainsci-11-01477-f001:**
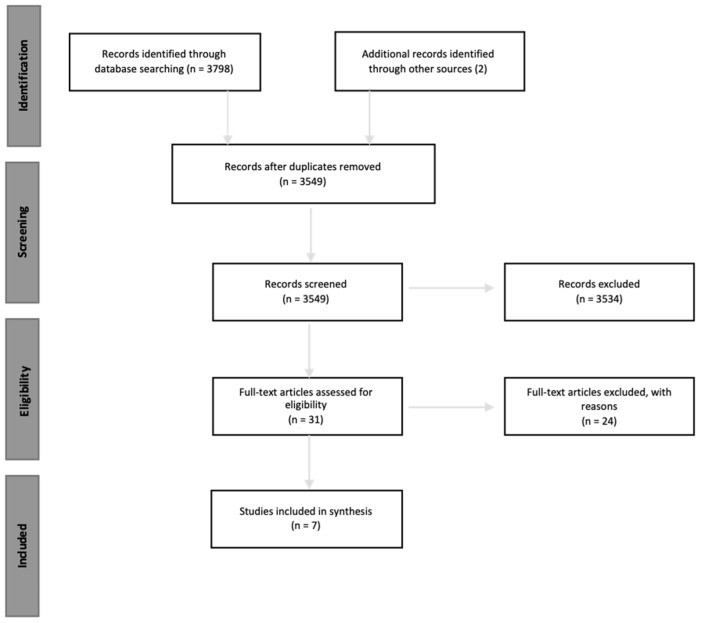
Flow chart for search strategy.

**Table 1 brainsci-11-01477-t001:** Search key terms.

Construct	Filter (Key Terms)
Physiological parameters	“SCL variability” OR “SCLV” OR “respiratory sinus arrhythmia” OR “RSA” OR “vagal tone” OR “cardiac slowing” OR “HRV” OR “heart rate variability” OR “SCL” OR “skin conductance level” OR “physiological measures” OR “physiological regulation” OR “physiological” OR “cardiac vagal tone” OR “cardiac vagal responses”
Attachment	“attachment theory” OR attachment OR “attachment style” OR “Insecure attachment” OR “Disorganized attachment” OR “Attachment model” OR “Attachment representations” OR “Internal working model” OR “IWM”
Psychopathology	“mental illness” OR “mental health” OR “mental disorder” OR “psychiatric disorder” OR disease OR symptom OR psychopathology OR problem OR disorder OR DSM OR ICD OR psychiatric
Mediation/ moderation analysis	mediation OR mediator OR mediating OR moderator OR moderating OR moderation OR “SEM” OR “structural equation modelling” OR “structural equation modelling” OR “(Baron and Kenny)” OR “indirect effect”

**Table 2 brainsci-11-01477-t002:** PICOS criteria.

	Inclusion Criteria
**Population**	All stages of human development (from early childhood to older adulthood), taken individually or as a dyad (mother-child or as a romantic couple)
**Intervention/** **exposure**	The presence of physiological processes as a mediator/moderator variable within the link between attachment (Independent variable) and different aspects of psychopathology (Dependent variable) such as disorders (e.g., specific DSM diagnosis), psychopathological indicators (e.g., anxiety/depression symptoms), and behavioural/psychological dysfunctions (e.g., emotional dysregulation, negative affectivity, peer-problems) consistent with theoretical evidence explored in the Introduction section
**Comparisons**	Control groups, same sample evaluated over time, differences between groups
**Outcomes**	Disorders (e.g., specific DSM diagnosis) Psychopathological indicators (e.g., anxiety, depression symptoms) Behavioral and psychological dysfunctions (e.g., emotional dysregulation, negative affectivity, peer-problems, etc.)
**Setting**	Any settings
**Study design**	Any study design (i.e., longitudinal or cross-sectional)
**Limits**	English and Italian language only; no temporal limits; only peer-reviewed studies
**Exclusion Criteria**	Attachment as moderator or mediator, positive outcomes (e.g., prosocial behaviour), no reference to disorders, or psychopathological indicators or behavioural/psychological dysfunctions, no measures of attachment, no physiological measurements, no empirical research, no peer-reviewed studies

**Table 3 brainsci-11-01477-t003:** Quality assessment of the included studies.

Author/s	Aim Clear	Design Appropriate to Aim	Sample Representative	Psychometric Characteristics	Acceptable Methods of Data Analysis	Significance of X on Mediator and of Mediator on Y	Interaction Test in Moderation	Clear Findings	Control Confounding Factors	Final Rating
Diamond et al. [68]	1	1	0	1	1	1	NA	1	1	7
Fagundes et al. [88]	1	1	1	1	1	NA	1	1	0	7
Conradt et al. [89]	1	1	1	1	1	NA	1	1	1	8
Sbarra et al. [90]	1	1	1	1	1	NA	1	1	1	8
Bosmans et al. [91]	1	1	1	1	1	NA	1	1	1	8
Murdock et al. [92]	1	1	0	1	1	1	NA	1	1	7
White et al. [93]	1	1	1	1	1	1	NA	1	1	8

NA = Not Applicable analysis.

**Table 4 brainsci-11-01477-t004:** Summary of reviewed studies.

Author/s (Year) Country	Sample	Design and Mediation/Moderation Analysis	Attachment Measures	Attachment Type and Attachment Figure	Physiological Measures	Outcome Measures	Statistical Findings	Quality Assessment Scale
Diamond et al. [68] US	75 young men (Mage = 21.69, SD = 3.47; 0% female)	CS; Baron & Kenny steps, Sobel test	ECR [94]; WHOTO [95,96]	Secure/Insecure; Current relationship	Baseline RSA (ECG plus latex rubber pneumatic bellows girth a sensor fitted around the participant’s chest)	Anger recovery (baseline minus post-task difference scores in RSA)	Adults with higher perception of security in their attachment relationships had more effective anger recovery and baseline RSA mediated this association (β = 0.18, *p* < 0.05; ΔR^2^ = 0.04, *p* < 0.10)	moderate
Fagundes et al. [88] US	110 teenagers at age 14 (50.9% female) and at age 18 (N = 71)	LO; Multiple linear regression	AAS-R [97]	Avoidance; Mother	RSA (ECG plus latex rubber pneumatic bellows girth sensor)	Modified versions of items from ETIG [98]	The interaction term between attachment avoidance and stress-induced changes in RSA is significant (β = −0.60, *p* < 0.001)Association between attachment avoidance and poor current adjustment to loss was positive for those with fewer stress-induced changes in RSA (*t* = 3.17, *p* < 0.001, r^2^ = 0.12)*,* whereas this association was negative for those with higher levels of stress-induced changes in RSA (*t* = −3.66, *p* < 0.001, r^2^ = 0.16)	moderate
* Conradt et al. [89] US	73 infants, age 5 months (Mage *=* 20.99 weeks, SD = 2.55; 55.8% female) and age 17 months (Mage = 17.6 months, SD = 1.76; 55.8% female)	LO; Hierarchical regression analysis	SSP [99]	Secure/insecure; Mother	Baseline RSA (bioamplifier)	BITSEA [100]	The interaction term between attachment relationship and baseline RSA is significant (β = 0.36, *p* = 0.01; ΔR^2^ = 0.09, *p* < 0.10). Specifically, there is a significant difference in problem behaviour depending on attachment classification only in infants with high baseline RSA (b = 7.25, *p* = 0.02; Disorganized: M = 31.25, SD = 16.68; Secure: M = 14.83, SD = 5.85)	strong
* Sbarra et al. [90] US	89 adults (Mage = 40.1, SD = 9.75; 65.2% female)	CS; Hierarchical regression analysis	ECR-R [101]	Avoidance; Close relationship	RSA (ECG plus respiratory effort transducer)	LOS; ROS [102]	Highly avoidant adults during the DMAT who had increase in RSA showed improvements in the self-concept reorganization three months later (*t* = 2.07, *p* = 0.03, r^2^ = 0.05), while highly avoidant adults who had a decrease in RSA showed relatively little self-concept reorganization (*t* = −2.10, *p* = 0.04, r^2^ = 0.05). The interaction term was significant (β = −0.19, *p* = 0.05; ΔR^2^ = 0.02) explaining 2% variability in the outcome variable	strong
Bosmans et al. [91] Belgium	60 children (Mage = 10.45, SD = 1.06; 50% female)	CS; Hierarchical multiple regression analysis	ECR-R children version [103]	Anxiety; Mother	SCL variability (electrodes on fingers in nondominant hand)	CES-D short form [104,105]	The interaction term between attachment anxiety and SCLV was significant (β = 0.39, *p* < 0.01; ΔR^2^ = 0.14, *p* < 0.01), suggesting that among children with higher SCLV there was a positive association between attachment anxiety and depressive symptoms (only in 34% of the children), but among those with low SCLV the association is negative	strong
Murdock [92] US	213 adults	CS; SEM Bootstrapping	ECR-SF [106]	Avoidance; Close relationship	Baseline RSA (ECG and respiratory band)	PSS [107]	Adult attachment avoidance was indirectly associated with self-reported health via general stress (β = −0.019, *p* < 0.05), but only among those with high stress-induced RSA (in 75.59% of the study sample) and not in adults with low stress induced RSA, demonstrating significant interaction between attachment avoidance and stress induced RSA (β = 0.17, *p* < 0.05)	strong
White et al. [93] Germany	165 children, school-age (Mage = 8.42, SD = 2.7 months; 46.1% female) of which 139 infants, preschoolers (Mage =5.19, SD = 5.9 months)	LO; SEM Bootstrapping	MSSB [108]	Positive and negative representations; Parents	C-S	SDQ—Peer Problems subscale; [109]	Positive representations at pre-school age predicted fewer peer problems at school age, through the path (β = 0.29, *p* < 0.01) from cardiac deceleration to peer problems (indirect effect: 95% CI [−1.416, −0.011])	strong

Legend—CS (Cross-Sectional); LO (Longitudinal); SEM (Structural Equation Modelling); ECR-R (Experience in Close Relationship Scale-Revised); ECR (Experience in Close Relationship Scale); SS (Strange Situation Procedure); MSSB (MacArthur Story Stem Battery); AAS-R (Adolescent Attachment Style Revised); SDQ (Strengths and Difficulties Questionnaire); CES-D (Center for Epidemiologic Studies Depression scale); BITSEA (Brief Infant-Toddler Social and Emotional Assessment); LOS (Loss of Self); ROS (Rediscovery of Self); ETIG (Expanded Texas Inventory of Grief); PSS (Perceived Stress Scale); RSA (Respiratory Sinus Arrhythmia); SCL (Skin Conductance Level); ECG (electrocardiogram); C-S (Cardiac slowing); HRV (Heart rate variability). * For the following papers (Fagundes et al. [88]; Sbarra et al. [90]) the authors also calculated the r-square results at the *t*-test for the single slope from the published data, as follows: r-squared = *t*-squared/(degrees of freedom + *t*-squared) (Rosenthal and Rosnow, 1991 [110]).

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
