# Peer review of "Physiological Factors Linking Insecure Attachment to Psychopathology: A Systematic Review"

_brainsci, 2021, doi:10.3390/brainsci11111477_

Round 1

Reviewer 1 Report

Thank you for the invitation to review this manuscript, which comments on an interesting/timely topic and utilizes appropriate methodology for a systematic review. The writing is generally good and clear (though many typos were found and proofreading is needed). Weaknesses include the conceptualization of the review’s aims and the loose association between that aim and what studies were actually reviewed. I’ve elaborated these points below. Finally, it is unclear what is learned from a review of such few studies with such varied characteristics. I encourage the authors to think about the take home message of their work and include that clearly in the abstract and discussion.

Major recommendations

  • More work should be done in the intro to link attachment to emotion regulation in order to justify the review’s focus on physiological measures that have been linked to emotion regulation.
  • Psychopathology as defined in the introduction varies from the actual outcome measures in the studies reviewed. This needs to be remedied. Please comment.
  • Introduction and discussion Focus on the aim of *developing* psychopathology, whereas the studies reviewed to not to be longitudinal or developmental. It seems in that sense that the aim of the review is not accomplished with the studies found. Please comment.

Minor points:

  • P. 1 “and appears to be prognostic of subsequent adaptation” should be appear
  • P. 2 “depending on the circumstances, environment, and individ- 61 ual differences.” Should add “the” before environment
  • P. 3 “ Its assessment uses the rapid action of the vagus nerve that overcome” should say overcomes
  • P. 3 “In developmental age, higher cardiac slowing is associated with peer re- 139 jection especially for female adolescents…” In what age?
  • P. 9 “Five studies on seven” should be of
  • P. 9 “The others three papers” should be other
  • P. 12 “Go further,” should be revised as this is not grammatically correct phrasing.

Reviewer 2 Report

This paper comprises a systematic review of the literature that examines the role of psychophysiological variables as mediators or moderators of the association between attachment variables and psychopathology (risk) variables. Seven papers were included in the review. Findings were summarized and studies were evaluated and discussed.

The paper has several strengths.  It examines an important question that is theory-driven.  The review includes papers written in two languages (English and Italian).  The paper also has some limitations and problems that I will outline below in hopes that they will help the authors improve the manuscript. 

The biggest problem I had with the paper was the fact that there was a lack of clarity regarding the definition and operationalization of the main outcome:  psychopathology.  Also, at times the authors state that they were examining psychopathology and other times they said psychopathology risk.  I became perplexed when I saw what words were used for the psychopathology search as I couldn’t understand why words like depressed/depression, anxiety, PTSD, OCD, etc. were not included.  The scope of the search is unclear, and because it isn’t specified well, then the list of words/terms included in the search may have been incomplete.  It would be legitimate if the authors wanted to limit themselves to nonclinical samples (for example, with high depressed mood) or to clinical samples only, or a combination of both.  Or if they wanted to focus on 1 or 2 diagnoses, that would be fine too.  The paper is lacking a clear justification of the search strategy.

The 7 studies included in the review have a kind of hodge-podge collection of outcome variables which makes it difficult to make sense of the literature.  Can the authors define their outcome variable and more clearly specify how that might be operationalized in the literature and choose search terms that follow from this?  That would make the paper more meaningful.  As is, I don’t have a good sense of what research might exist that is relevant but excluded due to the above problems (if any). 

Although the paper is generally clearly written, it is in need of some editing and checking for awkward phrasing and odd word usage (e.g., arranged, studio). 

The paper is very long considering it focused on only 7 studies.  Remove repetition.  Keep Discussion focused.

Describe the pattern of results found in studies fully (e.g., that examine SCL variability as a moderator; in what direction is avoidance associated with health? – poor health?).  Describe findings accurately. Weren’t the Bosmans findings about SCL variability, not SCL?

I don’t follow this sentence:  “This review analysis follows a narrative and thematic synthesis of qualitative research.”  You are examining quantitative research.

More minor points:

What is the principle of multi-finality?  Given this paper may have an audience of people with different areas of expertise, it is worth defining this.

Please refer to participants rather than subjects.

Footnote 3 has information that would be better placed in the Introduction.

Note – there are self-report measures of disorganized adult attachment.

Table 3 – what does the / indicate?

Table 4 – population should be sample.

There is an inconsistent reference style (e.g., capitalization of journal titles).

The page numbers restart.

I hope that the above comments will help the researchers to improve the presentation of their work as this is a potentially very interesting area.

Round 2

Reviewer 2 Report

The authors have taken into account my feedback from the first set of reviews to some extent.  The paper has improved, however, many of the key problems that I observed with the paper last time still remain. 

Again, the biggest problem is that the scope of the search is still unclear and specified poorly.  Importantly, there is a disconnect between the search terms used and the aims as specified in the Introduction.  The definitions of psychological constructs is still unclear and in places the authors seems again to conflate psychopathological risk factors and psychopathology outcomes or indicators.  This makes the paper confusing. 

The aims of the study need to be crystal clear and, most importantly, link up with the Method.  I still do not see how the method and choice of search terms fit with the aims of the systematic review.  What is your definition of psychopathological risk factors?  How does this fit with your search terms?  On page 2 you say, “In a recent review of a population of young people, Lynch et al. (2021) [13] highlighted that stressful life events, factors, peer and friendship problems, low effort control and negative affectivity among others are factors that increase transdiagnostic risk and are correlated with a general psychopathology factor (or 'p' factor, Caspi et al. 2014[14]) that seems to reflect a shared vulnerability between different mental disorders [15].”  I assumed that this was your definition of psychopathological risk factors.  This suggests that you might use search terms like stressful life events, or low effort control in your search for work on psychopathological risk factors but you do not.  Instead the search terms focus on things like mental illness and disorders.  You say you include diagnoses or disorders but don’t have terms for specific diagnoses.  I find the whole thing confusing.

Once we get to the Method and Results sections, the paper once again lumps together psychopathological outcomes and psychopathological risk factors and psychiatric indicators and outcomes.  These seem like different constructs.  They are not clearly defined in the paper.  See for example, in Table 2, the Intervention Box.  Table 1 only has a category called psychopathological risk factors but the key terms look like psychopathology outcomes (mental health).  But Table 2 talks about psychiatric indicators and outcomes together. 

In the Discussion section you talk about a symptom (depressive symptoms) being a risk factor.  A risk factor for what?  Isn’t a depressive symptom a psychopathological indicator or outcome? 

Although the writing has improved greatly from the editing, it still requires further editing and correction of typos to improve readability and clarity.  Some examples: Abstract line 22:  patterns should be pattern.  Page 2, line 71, outcomes has a typo.  Page 2, line 85 – the sentence doesn’t make sense. Page 2, line 92 – psychological risk or psychopathological risk?  Be precise.  Page 3, line 109 – fix sentence.  Page 5 – lines 218-221 - This sentence is unclear. Page 6, lines 254-7 – unclear sentence and rationale. Page 12, line 407- need a complete sentence.  Page 13, line 438 – unclear sentence.  Page 14, line 500-2 Paragraphs have more than one sentence.  Page 14, line 525, We defined psychopathology as or risk?  Unclear. Page 14, line 529 – unclear.

Other issues:  Page 2, line 77 attachment insecurity is (not may be) associated with mental health difficulties.

I hope the above comments help the authors to improve the paper.
